# Influence of Maternal Diet and Intergenerational Change in Diet Type on Ovarian and Adipose Tissue Morphology in Female Rat Offspring

**DOI:** 10.3390/medicina58070854

**Published:** 2022-06-26

**Authors:** Nenad Čekić, Anđela Grgić, Antonio Kokot, Robert Mujkić, Darija Šnajder Mujkić, Nikola Bijelić, Marko Sablić

**Affiliations:** 1Department of Anatomy and Neuroscience, Faculty of Medicine, Josip Juraj Strossmayer University of Osijek, 31000 Osijek, Croatia; nenad_cekic@hotmail.com (N.Č.); dsnajder@mefos.hr (D.Š.M.); sablicmarko1@gmail.com (M.S.); 2Department of Anatomy, Histology, Embryology, Pathological Anatomy and Pathological Histology, Faculty of Dental Medicine and Health, Josip Juraj Strossmayer University of Osijek, 31000 Osijek, Croatia; robert.mujkic@gmail.com; 3Department of Surgery, National Memorial Hospital Vukovar, 32000 Vukovar, Croatia; 4Clinical Institute of Nuclear Medicine, and Radiation Protection, University Hospital Osijek, 31000 Osijek, Croatia; 5Department of Histology and Embriology, Faculty of Medicine, Josip Juraj Strossmayer University of Osijek, 31000 Osijek, Croatia; nbijelic@mefos.hr

**Keywords:** high-fat diet, histomorphometry, epigenetics, ovaries

## Abstract

*Background and Objectives:* A high-fat diet causes inflammation in the organism and many metabolic disorders. Adipose tissue secretes adipokines that affect the function of many organs. The health status of the mother before and during pregnancy affects the health of the offspring. The aim of this study was to determine how the type of maternal diet and the change in the type of diet in the offspring affects the histological characteristics of the ovaries and subcutaneous and perigonadal adipose tissue in female rat offspring. *Materials and Methods:* Ten female rats were divided into two groups. One group was fed standard laboratory chow, and the other was fed a high-fat diet and mated with a male of the same breed. The offspring of both groups of dams were divided into four subgroups with different feeding protocols. At 22 weeks of age, the offspring were sacrificed. Ovaries and subcutaneous and perigonadal adipose tissue were isolated. In the ovaries, the presence of cystic formations was investigated. Histomorphometric analysis was performed in two types of adipose tissue. *Results:* The weight of the ovaries of the offspring of mothers fed a high-fat diet was significantly higher than that of the offspring of mothers fed standard laboratory diets. Cystic formations were found in the ovaries of the offspring of mothers fed a high-fat diet. In subcutaneous adipose tissue, the percentage of small-sized adipocytes was significantly higher in the offspring of mothers fed standard laboratory diets. There were no significant differences in adipocyte surface area and adipocyte number between groups. *Conclusion:* Maternal diet influences the morphology of the ovaries and adipose tissue of the offspring.

## 1. Introduction

In recent decades, obesity has become an epidemic in modern society [1], with visible consequences in both health and economic terms [2]. The main causes of obesity are increased dietary intake high in saturated fat and carbohydrates and a sedentary lifestyle. Obesity leads to the development of disorders and diseases of many body systems, including cardiovascular disease, diabetes, reproductive disorders, malignancies, depression, and other psychiatric disorders [3]. Interestingly, 10–30% of obese people are considered metabolically healthy, meaning that they have a lower risk of cardiovascular and metabolic disorders than metabolically unhealthy individuals [4]. Although caloric intake is more important than diet type in obesity, metabolic consequences depend on both caloric intake and diet type. For example, a recent study showed that a high-fat diet negatively affected the reproductive function of female rats regardless of caloric intake [5]. On the other hand, a high-fat diet with caloric restriction had a positive effect on plasma adipokine levels, glucose tolerance, lipid profile, and inflammation and oxidative stress in rats [6]. Other studies also confirm that the type of diet is one of the most important factors that can influence inflammation, metabolic responses, and behavior, including dietary habits [7,8,9,10].

Adipose tissue serves more than just the accumulation of fats. It physiologically influences the function of most organ systems and secretes various substances called adipokines. They are involved in the control of various processes, such as nutrition, insulin sensitivity, inflammation, etc. [11,12,13]. Based on the morphological characteristics, three main types of adipose tissue can be distinguished: white, brown, and beige adipose tissue. They are found in specific locations in the body. Depending on its location in the body, a distinction is made between subcutaneous (SAT) and visceral adipose tissue (VAT). VAT is thought to be involved in the development of several metabolic diseases. In laboratory animals, removal of VAT results in improvement of metabolic imbalance [14]. The volume of adipose tissue may increase by hyperplasia (an increase in the number of fat cells) or hypertrophy (an increase in the size of fat cells). Hyperplasia is referred to as a “healthy” increase in adipose tissue because adipocytes arise from progenitor cells. In contrast, adipose tissue hypertrophy is characterized by altered, dysfunctional adipocytes that are subject to apoptosis and sterile inflammation [15].

Maternal health status before and during pregnancy has a major influence on the health of the offspring both in the early perinatal period and later in life [16]. Psychosocial stressors and maternal nutrition potentially have the greatest influence on offspring health, either directly or through interactions with the genome or epigenome across generations [17]. Maternal high-fat diet during the preconception period causes oxidative stress in offspring and decrease in follicular growth and development [18]. One of possible explanations of ovarian disorders in offspring of mothers fed a high-fat diet is increased ovarian kisspeptin/GPR54 system [19]. Antioxidant supplementation in pregnancy improves oocyte quality and number in the offspring [20]. Maternal high-fat diet causes disturbances in insulin sensitivity and consequently adipocyte hypertrophy. This phenomenon is most visible in visceral adipose tissue of male offspring [12]. The influence of parental nutrition on offspring physiology or disease development has mostly been studied in animal models because of the methodological and ethical limitations of such studies in humans. Nevertheless, some characteristics of animal organisms limit the possibility of drawing conclusions about the human organism from animal studies. For example, the “prototype” of VAT in humans is omental adipose tissue (OAT), and the rat, which is an important animal model for obesity research, has no omental adipose tissue at all. The main depot of VAT in rats is the perigonadal adipose tissue (PAT) [14].

Several studies have shown the effect of adipose tissue dysfunction on the development of reproductive system disorders. For example, abdominal obesity is present in most women with polycystic ovary syndrome (PCOS) and affects both the metabolic and reproductive phenotypes of the syndrome [21]. Increased consumption of fatty foods leads to so-called silent inflammation. Silent inflammation is mediated in part by macrophages, which are found in various tissues and stimulate or suppress inflammatory processes depending on the environment [22]. In PCOS, an increased number of macrophages, especially their M1 fractions (proinflammatory macrophages), are found, and consequently the secretion of tumor necrosis factor (TNF-α) in the ovaries is also increased. The fraction of M2 macrophages (anti-inflammatory macrophages) is reduced [23].

Consumption of foods rich in saturated fatty acids affects the reproductive system [24]. This is interesting because reproductive physiology is stressed by the influence of circulating substances secreted from adipose tissue and substances from the organs and tissues of the reproductive system itself [5]. However, little is known about the influence of maternal diet and intergenerational changes in dietary form on the function and morphology of the female reproductive system. The objective of this study was to determine whether the type of maternal diet and the change in diet type in the next generation affect the histological characteristics of the ovaries and subcutaneous and perigonadal adipose tissue in female offspring of Sprague Dawley rats. The hypothesis of this study is that maternal high-fat diet causes the appearance of cystic formation in ovaries and a higher percent of adipose tissue hypertrophy in offspring.

## 2. Materials and Methods

### 2.1. Study Design

The first generation consisted of 10 female Sprague Dawley rats divided into two groups. One group was fed standard laboratory chow (Mucedolla, Settimo Milanese, Italy) for six weeks (CD group; *n* = 5), and the other group was fed a diet high in saturated fatty acids (HFD group; *n* = 5) (Table 1).

Females were mated with a male of the same breed. After birth and three weeks of lactation, the female offspring of both groups were divided into four groups of six female offspring each. Each group of offspring received a different feeding protocol (Figure 1): offspring from CD mothers fed standard laboratory diets (CD-CD), offspring from CD mothers fed diets rich in saturated fat (CD-HFD), offspring from HFD mothers fed standard laboratory diets (HFD-CD), and offspring from HFD mothers fed diets rich in saturated fat (HFD-HFD). Excess pups were removed.

Rats were bred in separate cages at an ambient temperature of 22 °C and a light–dark cycle of twelve hours. Standard laboratory chow and tap water were available ad libidum to the CD groups. The rats in HFD groups were offered with equal amounts of 30 g of a diet rich in saturated fat twice daily (at 9:00 a.m. and at 4:00 p.m.) to avoid increased caloric intake. The study was approved by the Ethics Committee of the Faculty of Medicine, Josip Juraj Strossmayer University of Osijek, and was conducted in accordance with the European Guidelines for Laboratory Animal Husbandry (Directive 2010/63/EU).

### 2.2. Tissue Preparation and Histological Examination

At the age of 22 weeks, the offspring were sacrificed. For this purpose, an injection of midazolam (Midazolam Torrex 5 mg/mL, Torrex, Chiesi Pharma, Parma, Italy) and ketamine (Ketanest S25 mg/mL, Pfizer, New York, NY, USA) was used according to the recommended doses. Ovaries and samples of SAT and PAT were isolated. The rats and ovaries were weighed (Beurer KS 22, Beijing, PRC). Relative weight of ovaries was calculated (ovary weight/rat weight). The tissues were stored in 4% formaldehyde for 3 weeks and embedded in paraffin blocks.

The paraffin blocks containing the ovaries were cut into 6-micrometer-thick sections using a Leica RM550 microtome (Leica, Vienna, Austria). The first and second sections were taken, and after nine discarded serial sections, the tenth section was taken, after which every tenth section was taken until the end of the tissue block. Staining was performed with hemalum and eosin. The obtained preparations were examined microscopically with an Olympus BX40 microscope and photographed with an Olympus C5050 camera (Olympus, Tokyo, Japan). The presence of cystic formations in the ovaries was investigated.

Paraffin blocks containing subcutaneous and perigonadal (around the ovary) adipose tissue were cut into 6-micrometer sections using a Leica RM550 microtome. The preparations were stained with hemalum and eosin and examined and photographed using the previously mentioned equipment. Digital images were captured and subsequently measured. CellProfiler v.2.1.1 (Anne E. Carpenter and Thouis (Ray) Jones, Cambridge, USA) free histomorphometry and cell phenotyping software, was used for histomorphometric analysis of adipose tissue. First, the marginal outlines of the cells on the slides were digitally separated. The adipocytes were then identified using the Primary Object Identification software module. Finally, the area of the adipocytes was measured using the Objects Size/Shape Measurement module and expressed in square micrometers. Adipocytes were classified into ten classes according to their surface area: (1) adipocytes with surface area < 2000 μm^2^, (2) 2000–2999 μm^2^, (3) 3000–3999 μm^2^, (4) 4000–4999 μm^2^, (5) 5000–5999 μm^2^, (6) 6000–6999 μm^2^, (7) 7000–7999 μm^2^, (8) 8000–8999 μm^2^, (9) 9000–10,000 μm^2^, and (10) > 10,000 μm^2^. The percentage of adipocytes in each class was calculated for each group of offspring and each adipose tissue type.

### 2.3. Statistical Analysis

All data were tested for normality using the Shapiro–Wilk test. Variables were expressed as median and interquartile range and standard deviation (Me [Q1–Q3]). The Mann–Whitney U test was used to test two independent groups, and the Kruskal–Wallis test was used to compare three or more independent samples. A difference was considered significant at *p* < 0.05.

## 3. Results

### 3.1. Ovary Weight

The highest weight and relative weight of ovaries was found in the group HFD-HFD. The lowest weight and relative weight of ovaries was in the CD-CD group, and it was significantly lower than in the HFD-HFD group and in the HFD-CD group. The weight of ovaries was significantly lower in the CD-HFD group than in the HFD-HFD group and in the HFD-HFD group. Relative weight of the ovaries was significantly lower in CD-HFD then in HFD-CD (Figure 2).

### 3.2. Ovarian Cysts

Cystic formations were found in ovaries of all offspring from the HFD-CD and HFD-HFD groups. No cystic formations were observed in any of the CD-CD and CD-HFD offspring. The difference is statistically significant, *p* < 0.01 (Figure 3).

### 3.3. Surface Area and Number of Adipocytes in SAT

The largest adipocytes were found in the HFD-CD group. The smallest adipocytes were found in the group CD-HFD. The highest number of adipocytes per unit area was found in the group CD-HFD and the lowest in the group HFD-CD. The differences in surface area and number of adipocytes were not statistically significant. At SAT, the percentage of small area adipocytes (0–2999) was significantly higher in groups CD-CD and CD-HFD than in groups HFD-CD and HFD-HFD (*p* = 0.02) (Figure 4).

### 3.4. Area and Number of Adipocytes in Perigonadal Adipose Tissue

The largest adipocytes were found in the group HFD-CD and the smallest in the group CD-CD. The highest number of adipocytes per unit area was found in the HFD-HFD group and the lowest in the HFD-CD group. The differences in surface area and number of adipocytes were not statistically significant. At PAT, the percentage of small area adipocytes (0–2999) was higher in the CD-CD and CD-HFD groups than in the HFD-CD and HFD-HFD groups. There was no statistically significant difference between the groups (Figure 5).

## 4. Discussion

The results of this study show that the type of maternal diet has an influence on ovarian weight and that ovarian cysts were found only in groups with significantly higher ovarian weight. The mothers of the offspring with higher ovarian weight and ovarian cysts were fed a high-fat diet. These groups of offspring also had disturbed alternation of reproductive cycle phases and abnormal cytological features of vaginal smears, as we have shown in previous studies. These groups also had significantly higher serum concentrations of TNF-α and IL-6 [25]. In the rat model of polycystic ovary syndrome, which is characterized by elevated levels of inflammatory markers, ovarian weight was increased [26]. These results are consistent with numerous studies that have shown that maternal nutrition has an important influence on the reproductive physiology of the offspring. This phenomenon is known as fetal programming or “developmental origin of health and disease hypothesis” [27]. In our study, offspring were given access to a high-fat diet twice daily in equal amount to prevent excessive food intake and resulting obesity, because we wanted to study the effects of diet type rather than obesity. Offspring did not develop obese phenotype and there was not statistically significant difference in body mass index between offspring [25]. Hussain et al. published a study claiming that a high-fat diet, regardless of caloric intake, has negative effects on reproductive physiology by disrupting the normal estrous cycle and promoting apoptosis of corpus luteum cells [5]. Kalem et al. have shown that an obese maternal diet results in fewer primordial follicles and thus a lower reproductive reserve of the ovaries [28]. Our previous studies have shown that mothers on a high-saturated-fat diet had a lower number of offspring, a lower percentage of successful conception, and that the offspring had irregular changes in the phases of the reproductive cycle, either in the form of prolonged cycles or the absence of some phases. Such cycles were very often anovulatory, so that conception was not possible [25]. A study conducted in rabbits showed the appearance of atretic follicles and cystic formations in the ovaries of the offspring of mothers on a high-fat diet [29]. Several putative mechanisms lead to these pathological changes, from oxidative stress to sterile inflammation in the periphery SAT or PAT, with consequent increased secretion of proinflammatory cytokines [18]. A study in pigs has demonstrated that a high-fat diet in the mother affects the ovarian health of the offspring by decreasing the number of healthy follicles and promoting oocyte apoptosis, increasing oxidative stress, and inducing a state of sterile ovarian inflammation [30]. Macrophages in the ovaries regulate cell proliferation, apoptosis, inflammation, and steroidogenesis, and are therefore the most important accessory cells for optimal reproductive function [31]. In further studies, it would be interesting to investigate the presence and distribution of macrophages in ovarian tissue depending on the type of diet.

At SAT, a greater adipocyte area was measured in the HFD-HFD and HFD-CD groups than in the CD-HFD and CD-CD groups. The highest number of adipocytes per unit area was calculated in the CD-HFD group and the lowest in the HFD-HFD group. In addition, the CD-HFD and CD-CD groups had a significantly higher proportion of small area adipocytes (0–2999 mm^2^). In the offspring of mothers who ate a high-fat diet, there was an increase in adipose tissue due to hypertrophy, although these offspring did not have a higher body weight. As mentioned earlier, hypertrophy is associated with inflammation and insulin resistance and consequently with various adverse health outcomes. These findings are also consistent with our previous study in which we showed that the offspring of mothers who ate a high-fat diet had higher serum concentrations of proinflammatory cytokines [25]. In addition, hypertrophic adipocytes are more prone to necrosis, which increases local inflammation in adipose tissue via serine phosphorylation of insulin receptor substrate-1 via nuclear factor κB and Jun N-terminal kinase signaling, leading to the development of insulin resistance [32]. Activation of immune system cells, particularly macrophages and T lymphocytes, is also enhanced [33]. Adipocyte hypertrophy increases local oxygen demand, leading to hypoxia and upregulation of proangiogenic factors and, consequently, local fibrosis, chronic inflammation, and a decrease in adiponectin secretion [34]. On the other hand, a significantly higher percentage of smaller surface adipocytes was found in the offspring of mothers fed a control diet. Thus, in these groups, adipose tissue expansion was mediated mainly by hyperplasia. During hyperplasia, adipocyte progenitor cells differentiate into adipocytes, a process mediated by several hormones and transcription factors. The most important of these is the peroxisome proliferator-activated γ-receptor (PPARγ) [35]. Koh et al. found that smaller adipocytes maintain a satisfactory level of insulin sensitivity and thus reduce the adverse metabolic effects of obesity. Such individuals are termed “metabolically healthy obese” [36]. Although not all differences between groups were statistically significant, it could still be concluded that maternal diet had a more significant effect on the model of adipose tissue proliferation than the diet of the offspring itself.

At VAT, as at SAT, larger adipocytes were found in the HFD-HFD and HFD-CD groups, whereas moresmaller adipocytes were found in the CD-CD and CD-HFD groups. Angiogenesis in the perigonadal VAT is more pronounced in female mice fed a high-fat diet than in male mice fed the same diet, and hyperplasia is more common in female perigonadal VAT. Thus, the negative effects of a high-fat diet on adipose tissue expansion in females may be less than in males; this should be further investigated in the future [37].

## 5. Conclusions

Regarding this research, it can be concluded that maternal high-fat diet influences the pathologic changes in ovary and adipose tissue morphology. The standard laboratory food of mothers was protective for offspring development. Nutritional interventions in one generation may set the stage for healthier offspring development, perhaps even more than the nutrition of the offspring itself.

## Figures and Tables

**Figure 1 medicina-58-00854-f001:**
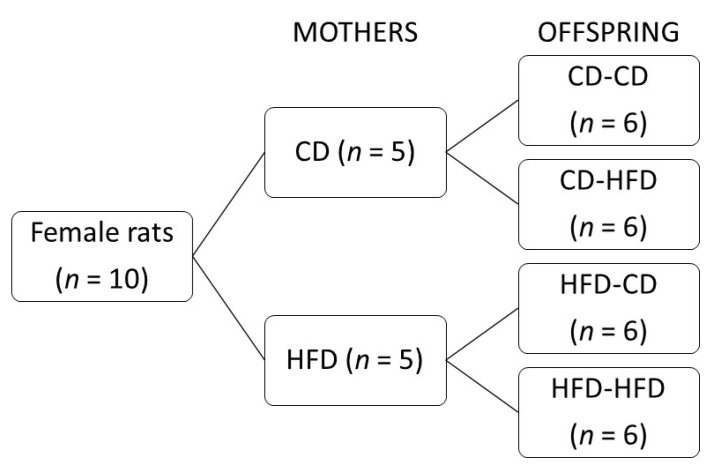
Study design.

**Figure 2 medicina-58-00854-f002:**
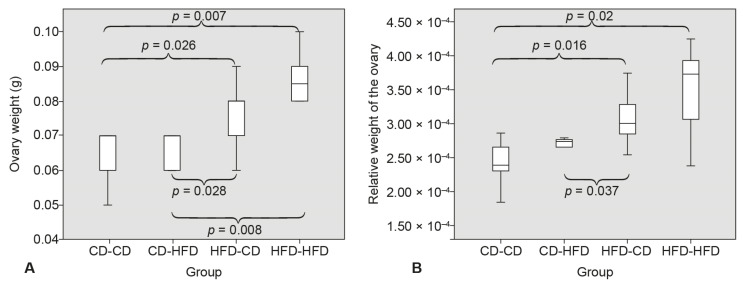
Ovary weight (**A**) and relative weight of the ovaries. (**B**) Mann–Whitney U test.

**Figure 3 medicina-58-00854-f003:**
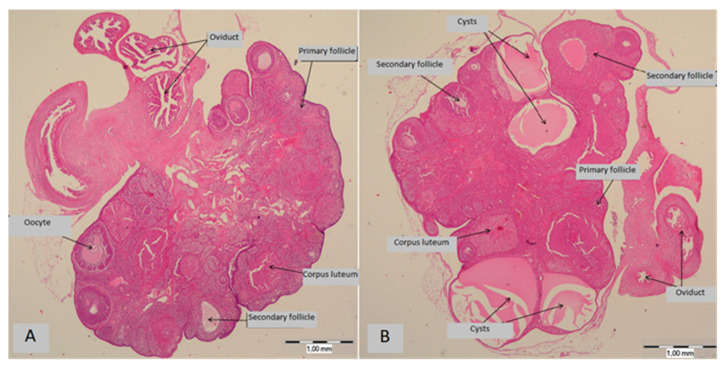
(**A**) Representative histological photography of normal ovary without cysts (groups CD-CD and CD-HFD). (**B**) Representative histological photography of ovary with cysts (groups HFD-CD and HFD-HFD). Magnification 20×.

**Figure 4 medicina-58-00854-f004:**
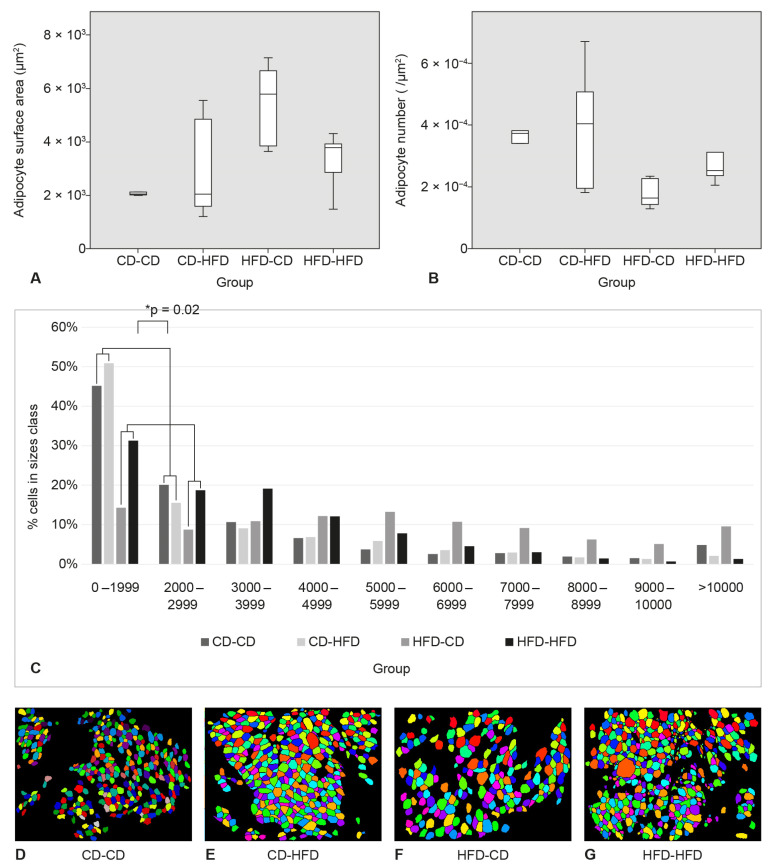
Adipocyte surface area, adipocyte number, adipocyte size classes, and representative images of adipocyte morphometry in subcutaneous adipose tissue (SAT). (**A**) Adipocyte surface area in SAT. (**B**) Adipocyte number per μm^2^ in SAT. (**C**) Adipocyte classes in SAT. (**D**–**G**) Representative images from four animals (one per each group) of adipocyte morphometry in CellProfiler in SAT.

**Figure 5 medicina-58-00854-f005:**
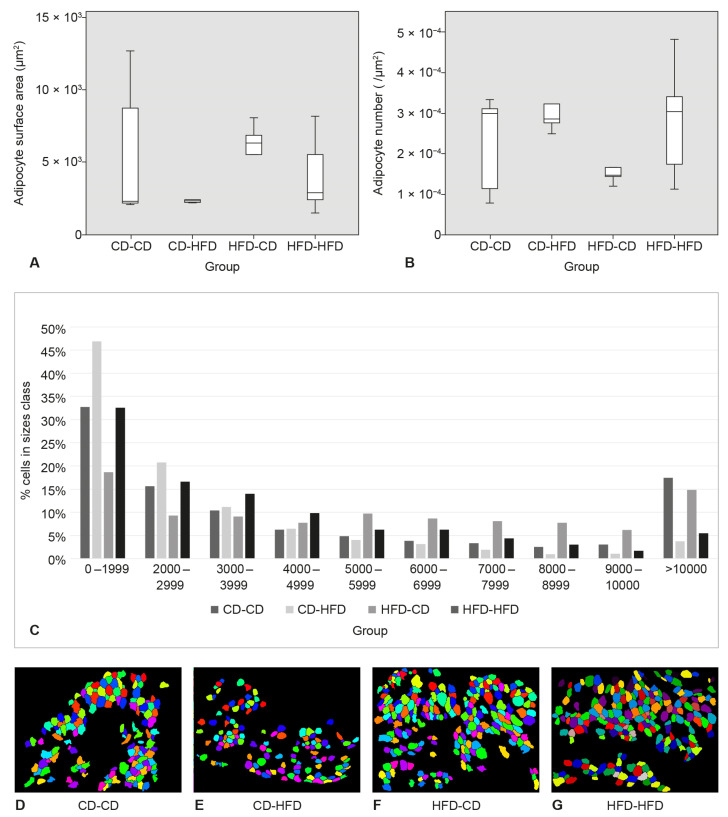
Adipocyte surface area, adipocyte number, adipocyte size classes, and representative images of adipocyte morphometry in perigonadal adipose tissue (PAT). (**A**) Adipocyte surface area in PAT. (**B**) Adipocyte number per μm^2^ in PAT. (**C**) Adipocyte classes in PAT. (**D**–**G**) Representative images from four animals (one per each group) of adipocyte morphometry in CellProfiler in PAT.

**Table 1 medicina-58-00854-t001:** Food components in control diet (CD) and high-fat diet (HFD).

Percent of Food Component (%)	CD		HFD
	Before 14 weeks of age	After 14 weeks of age	
Complex carbohydrates	53.7	66.5	30
Plant proteins	30.5	18.2	27.2
Animal proteins	4.7	3.5	
Fodder mix	3.0	7.5	9.7
Vitamins and minerals	4.1	3.2	3.9
Fats (soybean oil)	1.4	0.4	28
Amino acids	0.1	0.1	0.1

## Data Availability

Not applicable.

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
