# Peer review of "Influence of Maternal Diet and Intergenerational Change in Diet Type on Ovarian and Adipose Tissue Morphology in Female Rat Offspring"

_medicina, 2022, doi:10.3390/medicina58070854_

Round 1

Reviewer 1 Report

This study determined how the type of maternal diet and the change in the diet type of offspring affects the histological characteristics of the ovaries and adipose tissue in female rat offspring. The study design is clear, and the results and discussion are interesting and meaningful. I have a few questions listed here:

1. In the study design, each group has 5 dams (5 in the CD group, 5 in the HFD group), however, at weaning, six female offspring were divided into each subgroup. I think it is better to take one female offspring from each litter to avoid litter effect. Thus, you should have 5 female offspring in each subgroup.

2. The HFD were offered twice daily in the HFD groups to avoid increased caloric intake. So have you measured the energy intake (kcal) of offspring in each group? Are there any differences? And what is the metabolic phenotype of these offspring? Have you measured body weight and adipose weight? 

3. The results suggest that maternal diet rather than the diet of the offspring itself had a more significant effect on ovarian and adipose tissue morphology. I think using two-way ANOVA with maternal diet and offspring diet as two factors in statistical analysis is more suitable to draw this conclusion. 

Reviewer 2 Report

Dear authors,

Thank you for being able to read this manuscript. The idea is interesting, but some points need to be changed to improve readers' understanding.

1) Introduction

- The authors describe a lot about adipose tissue (brown, white…), but the types have not been studied. The same is true of inflammation. I believe this can be used in the discussion. On the other hand, the authors say very little about maternal alterations causing problems in the next generation. I would like that the intergenerational effects of a high-fat diet on ovaries and adipose tissue were better explored.

- There is no hypothesis. Please insert the hypothesis of this study in introduction.

2) Methodology

- The main focus of this study is diet. Therefore, I suggest the authors insert a table showing the constituents of the standard and high-fat chows.

- Studies using different generations are often confusing. The authors described the methodology well, but as a suggestion a figure of the experimental design could be made.

- In statistical section, I suggest remove the software used in the study. The result of the analysis must be the same regardless of the program used, or even if the calculations are done without any software. Therefore, I believe that this information is irrelevant. 

3) Results

- Why did the authors not use the relative weight of the ovaries? In addition, it would be interesting to present the weights of adipose tissues (subcutaneous and perigonadal).

- Figure 4 is about SAT?
